# OpenReview forum: "Optimizing Agent Planning for Security and Autonomy"
_ICLR.cc/2026/Conference — ICLR 2026 Poster_

### Official Review · Reviewer_VHQD · 2025-10-28

**Soundness:** 3
**Presentation:** 3
**Contribution:** 2
**Rating:** 2
**Confidence:** 3

**Summary:**

The paper discusses optimizing AI agent planning for security and autonomy by introducing autonomy metrics and proposing a security-aware agent called PRUDENTIA. PRUDENTIA improves autonomy by making the agent aware of security policies and planning for both task progress and policy compliance. The agent is designed to minimize human-in-the-loop (HITL) interventions (to generate security policies) while maintaining task completion rates. Experimental results show that PRUDENTIA shows noticeable improvement in security at a cost of having more interaction with human.

**Strengths:**

- The paper introduces new autonomy metrics (HITL load and TCR@k) that capture the benefits of deterministic defenses for AI agents, providing a more comprehensive evaluation framework. The metric is quite straightforward: it measures the proportion of tasks completed with k human interaction while avoiding security issues.
- The paper presents a thorough evaluation of PRUDENTIA on multiple benchmarks (AgentDojo and WASP), demonstrating its effectiveness in reducing HITL load and improving task completion rates across various models and scenarios.

**Weaknesses:**

- The paper assumes that a human will always be available to provide endorsements or approvals when necessary, which is unrealistic in many real-world deployments. Imho this assumption undermines the purpose of developing autonomous AI agents, as it relies heavily on human intervention. Moreover, human involvement not only introduces overhead but also the potential for human error. More critically, the emergence of more sophisticated attacks could exploit this human-in-the-loop (HITL) mechanism, potentially forcing the AI to conceal malicious activities from human oversight. For instance, advanced prompt injection attacks might be designed to manipulate the AI into presenting a sanitized version of its actions or intentions to the human reviewer, thereby bypassing the security checks.

- The paper's baseline comparison is limited to basic system-level defenses and lacks a comprehensive evaluation against more advanced system-level defenses, such as CaMeL. Additionally, it does not include a comparison against model-based defenses, including techniques like instruction hierarchy and StruQ. Incorporating these advanced defenses into the comparison would provide a more nuanced understanding of PRUDENTIA's relative strengths and weaknesses. Model-based defenses, in particular, offer an interesting counterpoint, as they aim to improve the robustness of LLMs against prompt injection attacks through different means and achieving TCR@0. A comparison against these approaches would help to clarify the advantages and disadvantages of PRUDENTIA's system-level defense strategy and identify potential areas for improvement or integration with other defense methodologies.

**Questions:**

- Is there any discussion on how one can tune PRUDENTIA's autonomy level and identify Pareto frontier?
- Does a human need to come up with initial set of security policies? Assuming if no initial policy is set, am I correct if I say that TCR@0 in this case will be equivalent to pure model-based defense (i.e. no human is involved)?

---

> ### Author Response · Authors · 2025-11-21
>
> > Experimental results show that PRUDENTIA shows noticeable improvement in security at a cost of having more interaction with human.
>
> Our results compare different agent designs, including Fides and Prudentia, *under the same security guarantees*. Rather than showing an improvement in security (which remains *constant*), our experiments show an improvement in autonomy. This is seen clearly in Figures 1 (middle and right) and 2, where Prudentia reduces HITL load and generally improves TCR@$k$ for low $k$.
>
> > The paper assumes that a human will always be available to provide endorsements or approvals when necessary, which is unrealistic in many real-world deployments. Imho this assumption undermines the purpose of developing autonomous AI agents, as it relies heavily on human intervention.
>
> This is a misunderstanding. TCR@0 represents task completion under **full autonomy**, and Figure 1 (right) shows that Prudentia improves over other approaches in this setting.
> Many real-world agents such as GitHub Copilot, Anthropic Computer Use, OpenAI Atlas and Perplexity Comet have user interfaces to ask for human intervention. Our insight is that in these deployments human intervention can be replaced by deterministic policy checks. We define HITL load as a metric to optimize for and show that agents that optimize for it, like Prudentia, push the boundary of what tasks can be completed securely with few or no human interventions compared to approaches that do not consider autonomy.
>
> > Moreover, human involvement not only introduces overhead but also the potential for human error. More critically, the emergence of more sophisticated attacks could exploit this human-in-the-loop (HITL) mechanism, potentially forcing the AI to conceal malicious activities from human oversight. For instance, advanced prompt injection attacks might be designed to manipulate the AI into presenting a sanitized version of its actions or intentions to the human reviewer, thereby bypassing the security checks.
>
> In Prudentia, HITL interventions trigger deterministically and cannot be manipulated by an attacker. We only use probabilistic LLMs to plan and choose variables to expand or endorse. This defends against bypassing HITL altogether. We clarify in Section 6 that we leave as future work the design of user interfaces for HITL prompts. Such interfaces should avoid presenting information in a manner that can be manipulated by an attacker, but we anticipate that IFC labels (which cannot be manipulated) and richer data provenance information could be used to present prompts that assist humans in making informed decisions.
>
> > The paper's baseline comparison is limited to basic system-level defenses and lacks a comprehensive evaluation against more advanced system-level defenses, such as CaMeL. Additionally, it does not include a comparison against model-based defenses, including techniques like instruction hierarchy and StruQ.
>
> We compare to Fides, a system-level defense on-par with CaMeL in terms of scope and guarantees. CaMeL uses a code-based planner and slightly different labels and policies, which make a fair comparison difficult. With these caveats, we offer to include an evaluation of CaMeL: we would instrument CaMeL to allow actions blocked by security policies (assuming a human approves them) and evaluate it on benign tasks in AgentDojo, extending Figures 1 and 2. Would the reviewer appreciate such an evaluation?
>
> Instruction hierarchy and StruQ are probabilistic defenses with no security guarantees, which is why we forgo a comparison. We refer to a recent pre-print (https://arxiv.org/abs/2510.09023) that shows that the combination of StruQ and instruction hierarchy can be bypassed with 100% ASR. This report also shows that 12 other probabilistic defenses can be bypassed as easily.

---

> > ### Author Response · Authors · 2025-11-21
> > **Answers to Reviewer's Questions**
> >
> > > Is there any discussion on how one can tune PRUDENTIA's autonomy level and identify Pareto frontier?
> >
> > Yes, in Section 5.1 we discuss the autonomy-utility tradeoffs of Prudentia compared to several baselines. It is possible that for different values of $k$ some planner designs achieve higher TCR@$k$ than others. Our experiments show that Prudentia is at or close to the Pareto frontier for low values of $k$ (see Figure 2) compared to the baselines we consider.
> >
> > > Does a human need to come up with initial set of security policies? Assuming if no initial policy is set, am I correct if I say that TCR@0 in this case will be equivalent to pure model-based defense (i.e. no human is involved)?
> >
> > Security policies would typically be specified by developers. For AgentDojo, we re-use the policies chosen by Fides while for WASP we choose the policies specified in Section 5.2. For a default initial policy that *allows all tool calls*, TCR@0 indeed corresponds to a fully autonomous agent that relies on the planner’s LLM alone to defend against prompt injection attacks, i.e., a pure model-based defense.
> >
> > Many real-world agents implement primitive policies. E.g., GitHub Copilot and Claude Code execute read-only tool calls without human intervention, whereas executing arbitrary shell commands requires human approval. Tool annotations in MCP (`openWorldHint`, `readOnlyHint`) can also be translated to policies with minimal effort. Prudentia uses policies based on IFC labels, which allow for more fine-grained control.

---

### Official Review · Reviewer_ueiJ · 2025-10-29

**Soundness:** 4
**Presentation:** 3
**Contribution:** 3
**Rating:** 6
**Confidence:** 3

**Summary:**

This paper presents a quantitative study on autonomy benefits of  deterministic system-level defenses in AI agents security. The paper introduces new autonomy metrics and evaluates a proposed security-aware agent PRUDENTIA on benchmarks. Results show the proposed approach archives higher autonomy without sacrificing task completion.

**Strengths:**

1. This paper presents the agent design of PRUDENTIA and deterministic system-level defense, which is well-motivated and aims to address a critical issue in agent security research.
2. To quantify the autonomy benefits and evaluate the proposed approach, this paper introduces new autonomy metrics and conducts extensive experiments.
3. This paper has good real-world applications, e.g., reducing security risks, improving user trusts, and safer integration with external data for AI agents.

**Weaknesses:**

1. This paper only focuses on indirect prompt injection attacks, the generalization of the proposed approach on other types of attacks is not discussed.
2. While the proposed approach outperforms baselines in autonomy, its completion rate shows some regression (73.2% vs. s FIDES’ 75.7%). This might suggest some minor trade-offs between autonomy and performance, but this is not discussed.

**Questions:**

1. For PRUDENTIA, what is the estimation method for the number of future PT calls? And what is the decision threshold for PRUDENTIA choosing endorsement?
2. Can you briefly discuss the generalization of the proposed approach in more open-ended environments? For example, how robust is the IFC-awareness of PRUDENTIA to complex unforeseen interactions?
3. What are the rough computation costs and latency introduced by the IFC mechanisms and PRUDENTIA’s planning?

---

> ### Author Response · Authors · 2025-11-21
>
> > This paper only focuses on indirect prompt injection attacks, the generalization of the proposed approach on other types of attacks is not discussed.
>
> Prudentia aims to defend primarily against indirect prompt injection attacks. Our threat model is the same as in Fides and CaMeL: the user, planner, and tools are trusted, whereas some external data sources are untrusted and controlled by the attacker. Therefore, attacks such as jailbreaks, direct prompt injection, and tool poisoning, where malicious instructions are embedded in a trusted source (e.g., user and system prompt, tool descriptions) are out of scope.
> Could you please elaborate on what other types of attacks you would like us to discuss?
>
> > While the proposed approach outperforms baselines in autonomy, its completion rate shows some regression (73.2% vs. FIDES’ 75.7%). This might suggest some minor trade-offs between autonomy and performance, but this is not discussed.
>
> We encourage the reviewer to consult Table 2 in the Appendix which presents the full results, including the standard error. The difference between 73.2%$\pm$5.2 and 75.7%$\pm$3.0 observed for TCR@$\infty$ with o4-mini is not statistically significant and is achieved under a 1.9x lower HITL load. Prudentia’s primary goal is to reduce HITL load and only secondarily improve TCR opportunistically.
>
> > 1. For PRUDENTIA, what is the estimation method for the number of future PT calls? And what is the decision threshold for PRUDENTIA choosing endorsement?
>
> We simply instruct the planner’s LLM to minimize the number of human interventions (endorsement or approvals). Although we do not specify an estimation method or decision threshold, we assign the same unit cost to endorsements and approvals, so that the LLM should only endorse a value when not endorsing it would lead to two or more HITL prompts to approve tool calls that depend on it. Please refer to Section 4 and Prudentia’s system prompt in Appendix D.3 for more details.
>
> > 2. Can you briefly discuss the generalization of the proposed approach in more open-ended environments? For example, how robust is the IFC-awareness of PRUDENTIA to complex unforeseen interactions?
>
> Our experiments on AgentDojo already show that Prudentia generalizes to open-ended environments. AgentDojo encompasses many tasks with complex unforeseen interactions, where the agent plans for dynamically retrieved data with unpredictable labels. We stress that, although we make the planner aware of per-tool policies, we do not specify how tool results are labelled. We observe that Prudentia explicitly plans ahead to account for this partial knowledge and defers certain decisions until the full information becomes available.
>
> > 3. What are the rough computation costs and latency introduced by the IFC mechanisms and PRUDENTIA’s planning?
>
> Below, we provide rough estimates for the costs of Basic-IFC, Fides, and Prudentia in terms of the number of input and output tokens and USD cents for solving AgentDojo tasks. We estimated this from logged traces rather than API responses, so we cannot give estimates for output tokens used by reasoning models. Roughly speaking, the overhead introduced by IFC mechanisms is reflected in the difference between Basic-IFC and Fides whereas planning for autonomy is factored in the difference between Fides and Prudentia.
>
> We believe that these costs can be brought down significantly as our goal was to investigate autonomy gains and we have not yet made a concerted effort to optimize costs. For instance, making calls to `plan` in parallel with other tools and (reinforcement) fine-tuning to bake into the model instructions and examples given in the system prompt can both lead to fewer turns and lower token counts.

---

> > ### Author Response · Authors · 2025-11-21
> > **Cost estimates**
> >
> > ## Mean input tokens
> >
> > | Model        | Method      | Banking            | Slack              | Travel             | Workspace          |
> > |:-------------|:------------|:-------------------|:-------------------|:-------------------|:-------------------|
> > | GPT-4o       | Basic-IFC   | 1023.77 ± 71.28    | 1342.03 ± 34.02    | 2879.62 ± 183.39   | 4273.48 ± 164.74   |
> > |              | Fides       | 7746.62 ± 729.44   | 11689.67 ± 447.49  | 16536.77 ± 679.04  | 8552.45 ± 605.68   |
> > |              | Prudentia   | 22577.69 ± 683.99  | 28368.55 ± 2650.24 | 37249.05 ± 2279.45 | 22423.03 ± 589.87  |
> > | o3-mini      | Basic-IFC   | 493.06 ± 61.26     | 1309.54 ± 209.20   | 1738.13 ± 294.53   | 2971.24 ± 613.10   |
> > |              | Fides       | 3538.65 ± 90.65    | 8468.39 ± 1234.11  | 12130.54 ± 1540.51 | 7629.85 ± 768.88   |
> > |              | Prudentia   | 17440.74 ± 926.52  | 27705.37 ± 1823.72 | 37575.25 ± 2824.56 | 24231.74 ± 561.60  |
> > | o4-mini      | Basic-IFC   | 792.39 ± 35.02     | 2040.68 ± 207.10   | 4289.18 ± 465.12   | 5451.01 ± 1314.89  |
> > |              | Fides       | 5367.80 ± 333.17   | 11634.30 ± 89.55   | 15551.39 ± 969.40  | 9753.03 ± 373.27   |
> > |              | Prudentia   | 31893.30 ± 3561.17 | 45162.51 ± 1831.12 | 55954.37 ± 970.29  | 32361.29 ± 1106.45 |
> >
> > ... of which the majority are cached
> >
> > | Model        | Method      | Banking            | Slack              | Travel             | Workspace          |
> > |:-------------|:------------|:-------------------|:-------------------|:-------------------|:-------------------|
> > | GPT-4o       | Basic-IFC   | 511.56 ± 63.52     | 852.46 ± 31.23     | 1784.70 ± 200.27   | 1992.31 ± 119.57   |
> > |              | Fides       | 5828.68 ± 700.04   | 9713.25 ± 411.14   | 13955.24 ± 659.41  | 6068.85 ± 529.77   |
> > |              | Prudentia   | 18567.28 ± 660.98  | 24352.77 ± 2582.42 | 32580.71 ± 2237.69 | 17829.85 ± 570.78  |
> > | o3-mini      | Basic-IFC   | 223.50 ± 37.79     | 928.86 ± 185.42    | 1185.91 ± 233.06   | 1593.58 ± 393.01   |
> > |              | Fides       | 1987.36 ± 72.49    | 6762.76 ± 1181.24  | 10003.99 ± 1458.04 | 5351.57 ± 669.54   |
> > |              | Prudentia   | 13570.00 ± 878.44  | 23787.00 ± 1773.53 | 33056.03 ± 2738.27 | 19690.05 ± 491.07  |
> > | o4-mini      | Basic-IFC   | 349.35 ± 27.79     | 1574.56 ± 179.14   | 3326.89 ± 403.29   | 3416.71 ± 1211.77  |
> > |              | Fides       | 3608.34 ± 309.77   | 9790.59 ± 84.28    | 13217.12 ± 941.40  | 7131.19 ± 312.39   |
> > |              | Prudentia   | 27393.17 ± 3444.05 | 40562.81 ± 1779.31 | 50554.95 ± 933.98  | 27271.16 ± 1023.11 |
> >
> > ## Mean output tokens
> >
> > | Model        | Method      | Banking         | Slack           | Travel          | Workspace      |
> > |:-------------|:------------|:----------------|:----------------|:----------------|:---------------|
> > | GPT-4o       | Basic-IFC   | 183.91 ± 7.46   | 284.82 ± 1.01   | 571.03 ± 6.69   | 217.72 ± 5.55  |
> > |              | Fides       | 333.51 ± 45.19  | 503.27 ± 28.65  | 886.96 ± 28.69  | 290.76 ± 11.56 |
> > |              | Prudentia   | 535.45 ± 24.33  | 672.86 ± 54.04  | 1049.62 ± 47.36 | 540.85 ± 17.29 |
> >
> > ## Cost (USD cents)
> >
> > | Model        | Method      | Banking        | Slack          | Travel          | Workspace     |
> > |:-------------|:------------|:---------------|:---------------|:---------------|:---------------|
> > | GPT-4o       | Basic-IFC   | ¢0.376 ± 0.026 | ¢0.514 ± 0.012 | ¢1.068 ± 0.073 | ¢1.037 ± 0.053 |
> > |              | Fides       | ¢1.542 ± 0.271 | ¢2.212 ± 0.163 | ¢3.277 ± 0.252 | ¢1.670 ± 0.212 |
> > |              | Prudentia   | ¢3.859 ± 0.253 | ¢4.721 ± 0.981 | ¢6.289 ± 0.847 | ¢3.918 ± 0.218 |

---

### Official Review · Reviewer_Qkcp · 2025-10-30

**Soundness:** 3
**Presentation:** 3
**Contribution:** 3
**Rating:** 6
**Confidence:** 4

**Summary:**

The main contribution of this paper is two metrics to evaluate the autonomy of AI agents with deterministic information flow control (IFC). IFC protects AI agents' integrity and confidentiality by adding restrictions on what data can be read by the agents and what actions can be executed by them. IFC deterministically protects agents from prompt injection attacks (PIAs) at the system level; however, it harms utility because it requires the user to approve read and execution actions when they violate the policy. This drawback is referred to as human-in-the-loop (HITL) approval. This paper introduces two novel and systematic metrics: HITL load and TCR@k. HITL load measures the total number of HITL events on tasks successfully completed. TCR@k measures the task completion rate under k HITLs. The authors also present Prudentia, a secure AI agent that generates plans with IFC-policy awareness. The authors introduce a novel endorsement mechanism. The user can endorse untrusted data to make it trusted, and the trusted label propagates to others to further reduce HITL.

**Strengths:**

1. The focused problem and proposed metrics are realistic. In the experience of using LLM-based AI agents, HITL approvals are disruptive and slow down the task completion process, yet they are necessary for security.
2. Policy and security label awareness encourage the agent to use safe and trusted tools and data. With this feature integrated into planning, the AI agent can reduce vulnerability at the root. This both reduces HITL interventions and avoids malicious actions.
3. The dual LLM design with strategic variable expansion allows the agent to choose to use the variables as a "pointer" or as data, which reduces the possibility of leakage in context.
4. The designs in Prudentia improve HITL load and TCR when k is low. In experiments, the authors also show that these designs do not harm the ultimate task completion (TCR@$\infty$).
5. Prudentia successfully defended all attacks and achieved a better task completion rate.

**Weaknesses:**

1. Evaluation. Both AgentDojo and WASP (based on VisualWebArena) contain different subtasks or websites. Previous work like CaMeL [1] also shows results on the subtasks. I suggest the authors include this analysis, since different tasks may require different steps and HITL interventions fundamentally, and averaging them out might not be reasonable.
2. Policy-aware planning needs additional analysis on the total number of steps and the time required to complete the tasks. The agents will try not to use sensitive or suspicious data and tools, which may lead them to take additional steps and time just to bypass these tools.
3. Comparison with the other methods (basic-IFC, Fides) is missing on the WASP benchmark. The authors should include HITL load and TCR@∞ on WASP as well.

[1]: Debenedetti et al., "Defeating Prompt Injections by Design." 2025

**Questions:**

Please address concerns in **Weaknesses**. And additionally:

1. Why does the TCR@$\infty$ of Prudentia not outperform Fides with o4-mini? Is this because the design of Prudentia limits some potential of the better models? The authors could include evaluation results of other models (GPT-5, Claude, Gemini) and show a correlation analysis of TCR@$\infty$ with the reasoning benchmarks.
2. If the DualLLM design already allows untrusted variables to propagate, what is the purpose of strategic variable expansion? The authors may include an analysis of each design in Prudentia to demonstrate their effectiveness and further enhance the soundness of the proposal.

---

> ### Author Response · Authors · 2025-11-21
>
> > Evaluation. Both AgentDojo and WASP (based on VisualWebArena) contain different subtasks or websites. Previous work like CaMeL also shows results on the subtasks. I suggest the authors include this analysis, since different tasks may require different steps and HITL interventions fundamentally, and averaging them out might not be reasonable.
>
> We provide a breakdown of results for each of the four AgentDojo environments in Figures 3, 4 and 5 in Appendix B. Figure 6 shows a similar breakdown of AgentDojo tasks grouped according to the taxonomy suggested by Costa et al. (2025). These more detailed results shed light on the benefits of policy-aware planners, e.g., highlighting their advantage over non-policy-aware planners in completing data-dependent tasks with few or no human interventions. We will provide a similar breakdown for WASP in a revision.
>
> > Policy-aware planning needs additional analysis on the total number of steps and the time required to complete the tasks. The agents will try not to use sensitive or suspicious data and tools, which may lead them to take additional steps and time just to bypass these tools.
>
> We did not keep logs of wall clock time, but we report below the mean number of turns on AgentDojo, which correlates well with time. There are 2 main reasons for additional turns: (1) Fides and Prudentia can use Quarantined LLM tool calls to process untrusted data; (2) Prudentia uses planning turns (calling the `plan` tool) to reason about variable expansion and endorsement. We will revise the paper to add this analysis, as well as a detailed breakdown of (1) cost in USD, (2) number of input tokens, (3) number of output tokens, (4) number of cached input tokens.
>
> | Base Model   | Algorithm   |   Banking |   Slack |   Travel |   Workspace |
> |:-------------|:------------|----------:|--------:|---------:|------------:|
> | GPT-4.1      | Basic-IFC   |     2.588 |   4.419 |    5.190 |       3.090 |
> |              | Fides       |     5.675 |   6.059 |   10.220 |       4.535 |
> |              | Prudentia   |     6.938 |   9.457 |   10.220 |       5.830 |
> | GPT-4o       | Basic-IFC   |     2.862 |   4.010 |    4.470 |       3.045 |
> |              | Fides       |     4.525 |   6.657 |    7.650 |       4.130 |
> |              | Prudentia   |     6.200 |   7.571 |    9.130 |       5.650 |
> | o3-mini      | Basic-IFC   |     1.950 |   3.914 |    3.710 |       3.110 |
> |              | Fides       |     2.362 |   5.210 |    6.550 |       3.900 |
> |              | Prudentia   |     4.875 |   7.438 |    9.396 |       5.945 |
> | o4-mini      | Basic-IFC   |     2.625 |   5.343 |    6.520 |       3.565 |
> |              | Fides       |     3.388 |   6.943 |    7.950 |       4.405 |
> |              | Prudentia   |     7.888 |  10.829 |   12.280 |       7.280 |
>
> > Comparison with the other methods (basic-IFC, Fides) is missing on the WASP benchmark. The authors should include HITL load and TCR@$\infty$ on WASP as well.
>
> We will add a comparison to Basic-IFC and Fides in a revision. Because all tasks in WASP are data-independent, there is no need for variable expansion or endorsement, and Fides performs similarly as Prudentia.

---

> > ### Author Response · Authors · 2025-11-21
> > **Answers to Reviewer's Questions**
> >
> > > 1. Why does the TCR@$\infty$ of Prudentia not outperform Fides with o4-mini? Is this because the design of Prudentia limits some potential of the better models? The authors could include evaluation results of other models (GPT-5, Claude, Gemini) and show a correlation analysis of TCR@$\infty$ with the reasoning benchmarks.
> >
> > Our results (see Figure 1 and Table 2 in Appendix B) indicate that Fides and Prudentia achieve similar TCR@$\infty$ with o4-mini (results are within the margin of error). This outcome is expected: Prudentia does not optimize for TCR@$\infty$, but rather minimizes the HITL load, for which Table 2 shows a clear improvement over Fides with o4-mini. More generally, our results show that Prudentia does not limit the potential of better models. On the contrary, it benefits from using newer generation reasoning models. We will be happy to include results for additional models in the revision.
> >
> > > 2. If the DualLLM design already allows untrusted variables to propagate, what is the purpose of strategic variable expansion? The authors may include an analysis of each design in Prudentia to demonstrate their effectiveness and further enhance the soundness of the proposal.
> >
> > Some data dependent tasks cannot be solved by simply propagating variables. The purpose of strategic variable expansion is to avoid expanding variables unnecessarily, which can result in increased HITL load and task failure. We use the `plan` tool to make the planner LLM think of better ways to use Quarantined LLM queries and complete tasks without directly accessing untrusted data. The effect of strategic variable expansion is evident when observing TCR@0 for data-independent tasks since tasks requiring a consequential tool call will fail after an unnecessary variable expansion due to the context being tainted. Figure 6 (top-left) in Appendix B shows that Prudentia achieves up to 25% higher TCR@0 compared to Fides by avoiding unnecessary variable expansions.
> >
> > The choice between endorsements and approval is another important component of Prudentia for reducing HITL load. Figure 6 (bottom-left) in Appendix B demonstrates the contribution of this component, with Prudentia reducing the HITL load by 2.5$\times$ for data-dependent tasks.
> >
> > If the reviewer believes a more detailed analysis strengthens our proposal, we can include selected figures and a discussion in the body of the paper.

---

> > > ### Comment · Reviewer_Qkcp · 2025-11-26
> > >
> > > I would like to thank the authors for their active engagement, detailed rebuttal, and additional experimental results.
> > > I have read through reviews by other reviewers and all the rebuttals. I remain positive about this paper.
> > >
> > > I have some follow-up questions about Table 2. It seems like GPT-4o with Basic and Basic-IFC constantly outperforms Fides and Prudentia on all TCR scores, and yet results on GPT-4o are not reported in the main text. I would like the authors to provide more analysis and discussion on this phenomenon. Does other related work also find GPT-4o a better base model than the reasoning models? Which of the designs makes Prudentia (and Fides) more suitable for reasoning LLMs?

---

> > > > ### Author Response · Authors · 2025-11-27
> > > > **Answer to follow-up questions**
> > > >
> > > > Thanks for these interesting questions. We will incorporate the analysis below in the main text.
> > > >
> > > > > Does other related work also find GPT-4o a better base model than the reasoning models?
> > > >
> > > > Yes, the difference between non-reasoning (GPT-4o) and reasoning (o3-mini) models can also be observed in related work. For instance, Debenedetti et al. (2025) report better baseline results with GPT-4o than o3-mini: 80.4 $\pm$ 7.9 vs 67.0 $\pm$ 9.4, but this seemingly reverses for CaMeL: 53.6 $\pm$ 9.9 vs 56.7 $\pm$ 9.9. The same can be observed in Costa et al. (2025) results, where an undefended baseline outperforms Fides with GPT-4o (74.5 $\pm$ 2.74 vs 60.00 $\pm$ 3.54) but the gap reverses and narrows for reasoning models (e.g., for o3, 91.00 $\pm$ 2.24 vs 93.50 $\pm$ 3.35).
> > > >
> > > > We present selected results with reasoning models in the main text to offer an evaluation that remains relevant as developers switch to them for agentic tasks. Somewhat unsurprisingly, [OpenAI's own cookbook](https://cookbook.openai.com/examples/reasoning_function_calls) recommends using reasoning models for "_multi-step planning for agentic workflows_", which describes well tasks in AgentDojo with the added complexity of IFC mechanisms in Fides and policy-awareness in Prudentia.
> > > >
> > > > > Which of the designs makes Prudentia (and Fides) more suitable for reasoning LLMs?
> > > >
> > > > We hypothesize that information hiding and extraction mechanisms (variable passing, quarantined LLM), asking models to avoid policy violations, and additional conversation turns, increase the complexity of tasks and demonstrate the superior planning capabilities unlocked by test-time computation in reasoning models.

---

### Official Review · Reviewer_V7nM · 2025-11-03

**Soundness:** 3
**Presentation:** 3
**Contribution:** 3
**Rating:** 6
**Confidence:** 3

**Summary:**

In this paper, they propose two metrics that take human interactions into account, which happens for real-world agents. They also introduce a policy-aware planning method to build a secure agent.

**Strengths:**

1. The research objective is interesting and bridge the gap between agents in research and agents in real world by taking human confirmation into account.

2. The paper is well organized, especially for the evaluation section, the findings and corresponding empirical evidence are clear. The models are also basically state-of-the-art models, which make the results more convincing.

**Weaknesses:**

1. It is not clear why the paper claims the agent can achieve security "guarantee". If I understand correctly, there may still be policy violation even with the policy aware planning.

2. The paper proposes metrics motivated by real agents but no evaluations on these agents are provided.

3. One key insight is that task completion rate is not a good metric for real world agents with human interactions, but no qualitative or quantitative analysis directly show how bad the task completion rate could be uncer certain conditions.

4. It seems be unclear why the defenses are categoried into probabilitics and deterministic defenses. What if LLM sin deterministic defense make a mistake (still a probablistic model in this sense), would it be not deterministic anymore?

**Questions:**

Please refer to the weakness part.

---

> ### Author Response · Authors · 2025-11-21
>
> > 1. It is not clear why the paper claims the agent can achieve security "guarantee". If I understand correctly, there may still be policy violations even with the policy aware planning.
>
> Prudentia's guarantee can be stated as: **every tool call satisfies a pre-specified policy or else is approved by a human**. Defenses against prompt injection attacks as implemented in Fides, which we build upon, and CaMeL guarantee adherence to a strict pre-specified policy: a tool call suggested by an agent is executed when it satisfies this policy or else is blocked. Prudentia relaxes a pre-specified policy with the ability for a human to approve a tool call when the policy is not satisfied, a situation we call a *policy violation*. Prudentia guarantees the absence of violations of this relaxed policy (a pre-specified policy with human approval fallback), while policy-aware planning reduces the need for human approval for actions that cannot be determined to comply with a pre-specified policy.
>
> > 2. The paper proposes metrics motivated by real agents but no evaluations on these agents are provided.
>
> Our evaluation is based on well-established prompt injection benchmarks (AgentDojo and WASP) to facilitate comparison to prior and concurrent work that uses these benchmarks, including Fides. We stress that these are representative benchmarks as they use ReAct-like planning loops similar to those used in real-world agents such as GitHub Copilot and OpenAI Atlas, they simulate realistic environments (e.g., browsing Reddit/GitLab websites for WASP, email/calendar APIs for AgentDojo), and that we evaluate on SoTA LLMs, meaning that our evaluation provides a close approximation to real-world deployments.
>
> > 3. One key insight is that task completion rate is not a good metric for real world agents with human interactions, but no qualitative or quantitative analysis directly show how bad the task completion rate could be under certain conditions.
>
> The key insight is that task completion rate alone is not a good basis for comparing agents with security guarantees against agents without guarantees because it does not show how they differ in terms of reliance on human oversight. Our evaluation on TCR@0 (see Figure 2) offers a glimpse at the capabilities of various agent designs to complete tasks fully autonomously on a level-playing field where deterministic security is table stakes. We refer to Fides (Costa et al., 2025) and CaMeL (Debenedetti et al., 2025) for additional qualitative and quantitative evaluations of planners with and without IFC-based defenses under benign conditions and under attack. Please let us know if there are other analyses that you would find useful to motivate this insight.
>
> > 4. It seems unclear why the defenses are categorized into probabilistic and deterministic defenses. What if LLMs in deterministic defense make a mistake (still a probabilistic model in this sense), would it not be deterministic anymore?
>
> Our primary goal is to defend against prompt injection attacks, which involve an agent taking an undesired action based on untrusted data. Probabilistic defenses such as those relying on *model alignment* and *content classifiers* reduce the likelihood of a successful attack but do not provide security guarantees. Deterministic defenses like Fides and CaMeL, on the other hand, enforce clear policies that can guarantee the absence of prompt injection attacks.
>
> The reviewer is right that neither probabilistic nor deterministic defenses per se protect against an LLM making a mistake and suggesting an inappropriate action when not influenced by an attacker, a point that we discuss briefly in Section 6. Deterministic defenses can be naturally extended to protect against LLM mistakes by enforcing stricter policies that assess actions even when they are not influenced by untrusted data (e.g., decisions about human subjects and financial transactions over certain thresholds may necessitate human oversight unconditionally). Some probabilistic defenses that track *task alignment* can also catch LLMs deviating from a given task by mistake (e.g., the *Tool filter* defense in AgentDojo removes the ability to call some tools altogether based on a probabilistic filter).

---

### Comment · Area_Chair_m2Am · 2025-11-22
**Official Comment by AC**

Dear Authors and Reviewers,

I would like to thank the authors for providing detailed rebuttal messages on time.

To reviewers: I would like to encourage you to carefully read all other reviews and the author responses and engage in an open exchange with the authors. Please post your first response as soon as possible within the discussion time window. Ideally, all reviewers will respond to the authors, so that the authors know their rebuttal has been read.

Best regards,
AC

---

### Comment · Area_Chair_m2Am · 2025-11-25

Dear Reviewers,

The authors have responded to your reviews. Please review and provide your feedback and responses.

Best,

Your AC

---

### Author Response · Authors · 2025-12-04
**Authors' summary of discussion with reviewers**

We thank the reviewers for their constructive feedback.

- Reviewer V7nM found that our work *“bridge[s] the gap between agents in research and agents in [the] real world”*, that it is *“well organized, especially for the evaluation section”*, its *“findings and corresponding empirical evidence are clear”*, and that our evaluation on *"state-of-the-art-models make[s] the results more convincing"*.
- Reviewer Qkcp appreciated that the “*focused problem and proposed metrics are realistic*”, our approach *“reduces HITL interventions and avoids malicious actions”*, and that it *“successfully defend[s] all attacks and achieve[s] a better task completion rate”*.
- Reviewer ueiJ found that the paper is *“well-motivated and aims to address a critical issue in agent security research”*, offering *“new autonomy metrics and extensive experiments”* with *“good real-world applications, e.g., reducing security risks, improving user trust, and safer integration with external data”*.
- Reviewer VHQD value our *“thorough evaluation of PRUDENTIA on multiple benchmarks […], demonstrating its effectiveness in reducing HITL load and improving task completion rates across various models and scenarios”*.

To address reviewers' questions and comments, we have posted individual responses. Additionally, we have revised our submission to include new results and clarifications based on the reviewers' suggestions. Please find below a summary of changes in our revision.

---

> ### Author Response · Authors · 2025-12-04
> **Summary of changes in the revision**
>
> ### 1. **Additional Experimental Results (Qkcp and ueiJ)**
>
> **Table 1 (WASP Benchmark Results):** We have added results including the number of turns for the WASP benchmark presented separately for the GitLab and Reddit scenarios. We added a discussion on the comparison to Basic-IFC and Fides baselines (Table 1). This addresses Reviewer Qkcp's request for a breakdown of our results on subtasks.
>
> **Table 2 (GPT-5 Results):** We have included results for an additional state-of-the-art reasoning model, GPT-5, across all methods and benchmarks (Table 2 in Appendix B). This provides broader evidence of our approach's effectiveness across different model capabilities, as requested by Reviewer Qkcp.
>
> **Cost and Efficiency Analysis:** We have added an analysis of computational costs in response to Reviewer Qkcp's request for "analysis on the total number of steps and time required to complete the tasks" and Reviewer ueiJ's question about "computation costs and latency." This includes:
> - Turns per task (by benchmark suite and model)
> - Token usage breakdown (input, cached tokens, and output tokens)
> - Cost estimates in USD.
> - Discussion of overhead introduced by IFC mechanisms and policy-aware planning
>
> This demonstrates that while Prudentia requires additional turns for planning, the overhead is reasonable and can be further optimized through techniques such as parallel tool calls and fine-tuning (see earlier response to Reviewer Qkcp).
>
> ### 2. **Clarification of Security Guarantees (V7nM)**
>
> **Introduction (Section 1, page 2):** We have added an statement clarifying our security guarantee model: "These systems guarantee that every tool call either satisfies the policy or is blocked (and can be escalated to human approval)." This directly addresses the anonymous reviewer's concern that "it is not clear why the paper claims the agent can achieve security 'guarantee'" and clarifies that policy violations with human approval still maintain deterministic security guarantees.
>
> ### 3. **Discussion of Model Selection: Reasoning vs Non-Reasoning Models (Qkcp)**
>
> **Evaluation (Section 5):** We have added a discussion explaining our focus on reasoning models, as they demonstrate superior performance for policy-aware planning tasks involving multi-step reasoning about IFC mechanisms. We note that Table 2 in Appendix B provides complete results that include non-reasoning models (GPT-4o) for completeness.
>
> ### 4. **Extended Discussion**
>
> We have significantly expanded our discussion (Section 6) to address other comments from the reviewers:
>
> **Components of Policy-Aware Planning:** This addresses Reviewer Qkcp's request: "If the DualLLM design already allows untrusted variables to propagate, what is the purpose of strategic variable expansion?" We provide detailed analysis showing:
> - Strategic variable expansion avoids unnecessary expansions for data-independent tasks, achieving up to 25% higher TCR@0
> - The choice between endorsements and approval reduces HITL load by $2.5\times$ for data-dependent tasks
> - These components work synergistically to improve autonomy (Figure 6 in Appendix B)
>
> **Scope of Defenses Considered:** This addresses Reviewer VHQD's concern about lack of comparison with probabilistic defenses. We clarify that:
> - Our evaluation focuses on deterministic defenses with security guarantees (Fides, on par with CaMeL)
> - Probabilistic defenses (StruQ, instruction hierarchy) provide no security guarantees. Recent work demonstrates these can be bypassed with 100% attack success rate (Nasr et al., 2025)
> - We compare defenses on a level playing field where security guarantees are table stakes
>
> **Threat Model and Attack Scope:** This addresses Reviewer ueiJ's question about "generalization of the proposed approach on other types of attacks." We explicitly state:
> - Our threat model targets indirect prompt injection attacks (same as Fides and CaMeL)
> - Out-of-scope attacks include jailbreaks, direct prompt injection, and tool poisoning
> - We focus on security guarantees and do not address general model errors (hallucinations) in benign scenarios
>
> **HITL Interface Design and Human Error:** This addresses Reviewer VHQD's concerns about "human error" and "sophisticated attacks exploiting HITL." We clarify that:
> - HITL triggers are deterministic and cannot be manipulated by attackers
> - Probabilistic LLMs only plan; IFC system determines approval decisions
> - Future work includes designing interfaces with IFC labels and data provenance to reduce human error
> - Such interfaces would display security context clearly to mitigate confirmation fatigue
>
> We believe that the revision addresses all concerns raised by reviewers. The core contributions—novel autonomy metrics and policy-aware planning—remain unchanged and are now better supported by additional experiments and clarifications. We remain committed to addressing any other concerns and welcome further feedback from the reviewers and the AC.

---

### Meta-Review · Area_Chair_xEzn · 2025-12-04

**Summary:**

The reviewers generally agree that the paper tackles an important problem, introduces useful autonomy metrics, and provides a well-executed evaluation of a security-aware agent. The main concerns raised were about clarity of the security guarantees, missing comparisons, the need for more task-level breakdowns, and discussion of overhead and trade-offs. The authors’ rebuttal addressed these points with detailed explanations and additional results.

Reviewer Qkcp explicitly reaffirmed a positive score after the rebuttal. Reviewers V7nM and ueiJ kept their marginal-accept scores and remained positive. Reviewer VHQD kept a lower score, though several of their concerns were clarified in the response.

After reading the paper and the rebuttals, I am convinced the work is solid and provides a meaningful contribution. I recommend accepting the paper, consistent with the majority of reviewers. I urge the reviewers to tune down the "guarantees" statements throughout.

**Reviewer Concerns:**

see above

**Reviewer Scores:**

see above

---

### Decision · Program_Chairs · 2026-01-26

Accept (Poster)